# Prevalence and factors associated with diagnosed diabetes mellitus among Asian Indian adults in the United States

**Ranjita Misra** [1]*, **Suresh S. Madhavan**[2], **Trupti Dhumal**[3], **Usha Sambamoorthi**[2]

**1** Department of Social and Behavioral Sciences, School of Public Health, West Virginia University, Morgantown, West Virginia, United States of America, **2** College of Pharmacy, University of North Texas System, Denton, Texas, United States of America, **3** Department of Pharmaceutical Systems and Policy, School of Pharmacy, West Virginia University, Morgantown, West Virginia, United States of America

* ramisra@hsc.wvu.edu

## Abstract

Higher prevalence of diabetes mellitus (DM) has been documented among South Asians living in the United States. However, combining the south Asian subgroups into one category masks the heterogeneity in the diagnosed DM, after controlling for known protective and risk factors. We assessed the association of Asian Indian ethnicity to diagnosed DM using a nationally representative sample of 1,986 Asian Indian adults in the US compared to 109,072 Non-Hispanic Whites (NHWs) using disaggregated data from the National Health Interview Survey (2012–2016) (NHIS). 2010 US census figures were used for age-sex standardization. Age-sex adjusted prevalence of DM was 8.3% in Asian Indians as compared to 5.8% in NHW. In adjusted multivariable logistic regression models, Asian Indians had higher odds ratios of reporting diagnosed DM compared to NHWs (AOR = 1.39, 95% CI: 1.12, 1.71). This association remained strong and significant even after controlling for other risk factors in the model (AOR = 1.47, 95% CI: 1.16, 1.85). Results suggest a favorable socioeconomic profile of Asian Indians was not protective on diagnosed DM. In addition, they were more likely to have diagnosed DM due to higher prevalence of obesity despite healthier behaviors of smoking and exercise.

## Introduction

The world-wide prevalence of diabetes mellitus (DM), a disabling chronic condition, is growing and is projected to increase to 700 million by 2045 [1]. South Asian countries (India, Bangladesh, Pakistan, Sri Lanka, Bhutan, Nepal and Maldives), and China account for approximately 60% of the world's population with diabetes [2, 3]. DM prevalence in these countries has increased more than 2.5 fold during the last decade and is expected to increase exponentially with time [3]. Migrant Asian Indians in the U.S. have high rates of insulin resistance due to an inherent genetic predisposition, and an increased disease incidence at lower age and body mass index (BMI) [4, 5] even with a favorable socio-economic profile (i.e. high income and high levels of education). Hence, migrant Asian Indians in the U.S. are reported to have

**Data Availability Statement:** The data we used for this paper were acquired from the National Health Interview Survey (https://www.cdc.gov/nchs/nhis/data-questionnaires-documentation.htm) and can

be freely downloaded by any researcher. This study had no special privileges. Inclusion criteria specified in the Methods section would allow other researchers to identify the study cohort and replicate all our analyses.

**Funding:** The authors received no specific funding for this work.

**Competing interests:** The authors have declared that no competing interests exist.

the highest ethnic-specific DM rates and higher risk of DM as compared to other ethnic groups [6–11]. Findings from a 20-year longitudinal study from the United Kingdom suggest that DM prevalence is three times higher among Asian Indians compared to the European controls [12]. Similar high prevalence of DM among immigrant Indians is reported in Singapore as compared to the native population [13].

Asian Indians (or Indian Americans) encompass 4.4 million people in the United States [14]. They are the second largest and fastest growing Asian subgroup in the US with a growth rate of 70% between 2000–2010 and expected to almost triple by 2050 [15]. Published studies have focused on DM in South Asians due to of small sample sizes and lack of disaggregated national data, potentially masking the disease burden within the subgroups [16]. Using National Health and Nutrition Examination Survey (NHANES) data from 2011 to 2016, Cheng et. al. reported that the prevalence of age-sex adjusted diagnosed DM among south Asians was higher (16.0% vs 8.2%) than prevalence among NHWs [17]. The heterogeneity of South Asians living in the US in terms of ethnic, religious, linguistic, and socioeconomic characteristics is a factor is masking disease burden within the subgroups. Hence, disaggregating the data has clear advantages for examining epidemiological trends over time, reduce health disparities, and understanding the DM prevalence in various racial/ ethnic groups. DM is increasing faster among Asian Americans subgroups than non-Hispanic Whites, non-Hispanic Blacks, and Hispanic Americans [18–20].

Despite a surge in investigation of chronic diseases in Asian Americans and its subgroups, population-based studies on DM prevalence among migrant Asian Indians in the United States are limited due to small sample size, use of purposive sampling and/or specific subgroups in various regions that limits the generalizability of the results, increase measurement errors in risk factors, or lack a comparison group [21–24]. For example, a 2004 community-based survey that included only Asian Indians living in the metro area of Georgia, Atlanta, found that 18.3% had DM [25]. Based on National Health Interview Survey (NHIS) data from 1997 through 2005, Asian Indians were three times as likely as NHWs to report DM even after adjusting for variables like age, sex, and obesity [26]. Using cross-sectional data from the NHIS data from 1997 through 2000, Mohanty et al., concluded that despite lower-rates of obesity, Asian Indians were more likely to have diagnosed diabetes compared to NHWs after controlling for age and obesity [21]. However, this study did not use the appropriate BMI for Asian populations as recommended by the World Health Organization (WHO) [27]. The Diabetes among Indian Americans (DIA) study estimated the prevalence of diabetes among U.S. Asian Indians residing in 7 US cities at 17.4% [22].

These studies have suggested a growing burden of DM in Asian Indians that represent a public health challenge. An examination of the prevalence of diagnosed DM and associated protective and risk factors among Asian Indians can assist in targeted prevention and treatment efforts in this ethnic group. As Asian Indian population is expected to increase, almost triple its size by 2050 [28], examination of diabetes prevalence in this group is important. Therefore, the primary objective of this study is to evaluate the association of Asian Indian ethnicity to diagnosed diabetes, after controlling for known protective and risk factors, as compared to non-Hispanic whites (NHWs) in the United States. We will use disaggregated data for Asian Indians from the NHIS, a nationally representative sample of non-institutionalized civilian population of households.

## Methods

The Institutional Review Board of West Virginia University and the University of North Texas Health Sciences Center determined that the study was exempt from IRB approval because the study used publicly available data.

## Study design

A cross-sectional study design was adopted using pooled secondary data from multiple years (2012–2016) of the NHIS. Data was pooled to ensure adequate cell size and to increase reliability of results. NHIS investigators recommend pooling multiple years of data to achieve adequate sample size and minimize the relative standard error (less than 30%) [29]. We included only Asian Indians and NHWs as our primary interest was to compare these two groups.

## Data source

NHIS is an ongoing, continuous, nationwide, cross-sectional annual survey of household civilian noninstitutionalized population in the US [30]. The nationally representative data is obtained by utilizing a multistage sampling technique, wherein the target universe is divided into numerous nested levels of strata and clusters [30]. This study used data from family, person, sample adult core, and imputed income files. The person file captures attributes such as sociodemographic characteristics, health status, and health insurance, whereas, information on the poverty status of the household was provided by the family files. Information about chronic physical conditions, psychological distress, access to care, and utilization of healthcare services were utilized from the sample adult core files [31].

## Study sample

The study sample consisted of NHIS participants who were 18 years or older, who were either Asian Indian or NHWs, and participated in the sample adult core. As our primary interest is comparing Asian Indians and NHWs, other racial/ethnic groups were excluded from the analysis. The final sample consisted of 111,058 adults (1,986 Asian Indians and 109,072 NHWs).

## Measures

**Dependent variable: Self-reported diagnosed diabetes—Yes/No.**   To ascertain an individual's diabetes status, a positive response to the following question was used: "Have you ever been told by a doctor or health professional (other than during pregnancy, if female) that you have diabetes?". Those who answered border-line DM were considered as not having diabetes. Also, pregnant women with gestational diabetes were excluded.

**Key independent variable: Asian Indians versus non-Hispanic Whites.**   Data on Asian American subgroups have been collected by the NHIS since 1992 but desegregated data for Asian Indians was only available since 2011 [32]. NHIS asks a number of queries to ascertain an individual's race or ethnicity. We used responses from a question that queried whether the respondent is 1) Hispanic, Latino/a, or Spanish origin, and 2) what his/her race is. Individuals responding, "not of Hispanic, Latino and/or Spanish origin" and "White" were categorized as NHWs. Asian subgroups were designated to seven subcategories: Asian Indian, Chinese, Filipino, Japanese, Korean, Vietnamese, and other Asian. For the purpose of this study, we only included individuals who self-identified themselves as Asian Indians.

**Demographic and lifestyle characteristics.**   We included variables that are known to be associated with DM risk among adults. These comprised of biological variables such as age (18–44 years, 45–64, and 65 or older) and sex (women/men). Socio-economic factors included education (less than high school, high school/GED, some college, and college), federal poverty level (FPL) ($< 100\%$, 100 to $< 200\%$, 200 to $< 400\%$ and $> = 400\%$) and employment (employed vs not employed). Access to healthcare (health insurance) defined as having health insure and no health insurance. General health was self-reported on a five-point Likert scale, ranging from excellent to poor and diagnosis of chronic conditions such as diabetes, hypertension, high

cholesterol, COPD, and heart disease by a doctor or health professional as yes/no. Lifestyle health practices included smoking status (nonsmoker, former smoker, and current smoker), alcohol use (lifetime abstainer, former drinker, and current drinker), physical activity (daily, weekly, monthly/yearly, and unable), and body mass index (BMI). Obesity was assessed by BMI cutoffs using both standard criteria and the World Health Organization Western Pacific Region (WHO-WPR) (World Health Organization Western Pacific Region, International Association for the Study of Obesity, International Obesity Task Force, 2000). The Centers for Disease Control's BMI criteria was used to classify NHWs as follows: 1) Underweight/normal (0–25.0 kg/m2); 2) Overweight (25.0–30.0 kg/m2); and 3) Obese ($\geq$ 30 kg/m2). For Asian Indians, we used both the standard and WHO recommended revised cut points for Asians that defined overweight as a BMI of 23.0–24.9 and obesity as a BMI$\geq$25 [33–35]. For all variables with missing values, we included indicator variables representing missing group in adjusted analyses. Self-reported general health (excellent/very good, good, fair/poor) and high cholesterol (people who answered "yes" to the question "have you ever been told by a doctor or other health professional that you had high cholesterol?"). Health insurance coverage (defined as insured and uninsured) and have usual source of care.

## Statistical analysis

To account for the complex sampling design, clustering, stratification, and weight variables were used, and all statistical analyses were conducted using SAS 9.4 (SAS institute INC, Cary, North Carolina) survey procedures. Unadjusted group differences in sample characteristics between Asian Indians and NHWs were analyzed using Rao-Scott chi-square tests. To derive age-sex adjusted prevalence of DM, we used 2010 US census distribution of age and sex [36]. Multivariable logistic regressions were used to analyze the associations of Asian Indians ethnicity to diagnosed DM, adjusted for known factors associated with DM—age, sex, education, employment, poverty status, health insurance, marital status, hypertension, high cholesterol, physical activity and smoking. Statistical inferences were based on a significance level of P (two-sided) $\leq$ 0.05.

## Results

Study sample consisted of 109,072 NHWs (98.2%) and 1,986 Asian Indians (1.7%) (Table 1). A little over half (51.1%) of the participants were female (49.5% AIs and 51.5% NHWs) and younger than 65 years of age (78% total, 92% AIs and 77.7% NHWs). A quarter (27.8%) were obese (9.8% AIs and 27.4% NHWs) according to the standard CDC criteria (BMI $\geq$30.0). In addition, the World Health Organization obesity classification for South Asians was used to calculate obesity for Asian Indians ($\geq$25.0). Using the WHO criteria, obesity rate was significantly higher i.e., 46.1% for AIs. Current smoking status was reported by only 17.9% of the participants. In general, over a quarter of the participants (29.9%) and 2/3rd (68.3%) reported a diagnosis of hyperlipidemia and hypertension, respectively.

We observed statistically significant subgroup differences among Asian Indians and NHWs except for sex and health insurance (Table 1). Asian Indians were more educated (highest percentage of college education) (72.9%) as compared to NHWs (34.4%). Moreover, Asian Indians were less likely to be current smokers (5.1% vs. 18.1%), more likely to engage in weekly physical activity (40.6% vs 37.2%), less likely to report hypertension (18% vs 31.8%), and more likely to be obese (46.1% vs 27.4%) compared to NHWs. World Health Organization obesity classification for South Asians was used to calculate obesity for Asian Indians ($\geq$25.0) and standard criteria was used for NHWs ($\geq$30.0).

**Table 1. Characteristics of Asian Indian and non-Hispanic White adults (n = 111,058), National Health Interview Survey, 2012–2016.**

| ALL | Total | | Asian Indians | | Non-Hispanic Whites | | Chi-sq | Prob [b] |
|---|---|---|---|---|---|---|---|---|
| | N | Wt % | N | Wt % | N | Wt % | | |
| | 111,058 | 100.0 | 1,986 | 100.0 | 109,072 | 100.0 | | |
| **Diabetes Mellitus** | | | | | | | 2.8 | 0.095 |
| Yes | 10,671 | 8.8 | 137 | 7.5 | 10,534 | 8.8 | | |
| No | 100,387 | 91.2 | 1,849 | 92.5 | 98,538 | 91.2 | | |
| **Sex** | | | | | | | 1.8 | 0.186 |
| Women | 60,020 | 51.5 | 914 | 49.5 | 59,106 | 51.5 | | |
| Men | 51,038 | 48.5 | 1,072 | 50.5 | 49,966 | 48.5 | | |
| **Age Groups** | | | | | | | 265.2 | < 0.001 |
| 18–39 | 33,775 | 33.9 | 1,186 | 53.3 | 32,589 | 33.5 | | |
| 40–49 | 16,342 | 16.4 | 364 | 21.3 | 15,978 | 16.3 | | |
| 50–64 | 30,501 | 27.7 | 277 | 17.3 | 30,224 | 28.0 | | |
| >= 65 | 30,440 | 22.0 | 159 | 8.0 | 30,281 | 22.3 | | |
| **Marital status** | | | | | | | 182.0 | <0.001 |
| Married | 58,718 | 52.8 | 1,358 | 68.3 | 57,360 | 52.5 | | |
| Wid/Sep/Div | 30,496 | 27.4 | 142 | 7.15 | 30,354 | 27.8 | | |
| Never married | 21,615 | 19.4 | 483 | 24.3 | 21.132 | 19.3 | | |
| **Education** | | | | | | | 771.6 | < 0.001 |
| LT HS | 9,855 | 8.5 | 88 | 5.2 | 9,767 | 8.5 | | |
| HS | 27,661 | 24.9 | 175 | 10.1 | 27,486 | 25.2 | | |
| Some College | 35,645 | 31.8 | 207 | 11.4 | 35,438 | 32.2 | | |
| College | 37,587 | 34.5 | 1508 | 72.9 | 36,079 | 33.8 | | |
| **Poverty Status** | | | | | | | 60.4 | < 0.001 |
| < 100% FPL | 12,239 | 8.5 | 219 | 8.3 | 12,020 | 8.5 | | |
| 100-<200% | 17,590 | 13.8 | 217 | 10.3 | 17,373 | 13.8 | | |
| 200-<400% | 30,177 | 26.9 | 379 | 20.6 | 29,798 | 27.0 | | |
| >= 400% | 41,616 | 42.1 | 1,003 | 52.6 | 40,613 | 41.9 | | |
| **Health Insurance** | | | | | | | 0.3 | 0.845 |
| Yes | 100,731 | 90.7 | 1,804 | 90.7 | 98,927 | 90.7 | | |
| No | 10,013 | 8.9 | 175 | 9.0 | 9,838 | 8.9 | | |
| **Hypertension** | | | | | | | 115.5 | < 0.001 |
| Hypertension | 38,098 | 31.6 | 320 | 18.0 | 37,778 | 31.8 | | |
| No hypertension | 72,861 | 68.3 | 1,665 | 82.0 | 71,196 | 68.1 | | |
| **High Cholesterol** | | | | | | | 51.7 | <0.001 |
| Yes | 33,291 | 29.9 | 380 | 19.1 | 32,911 | 30.1 | | |
| No | 77,347 | 69.6 | 1,602 | 80.7 | 75,745 | 69.4 | | |
| **Body Mass Index** | | | | | | | 198.9 | < 0.001 |
| Und/normal | 39,525 | 35.6 | 650 | 30.5 | 38,875 | 35.7 | | |
| Overweight | 36,710 | 33.0 | 415 | 21.2 | 36,295 | 33.3 | | |
| Obese [a] | 30,940 | 27.8 | 882 | 46.1 | 30,058 | 27.4 | | |
| **Smoking Status** | | | | | | | 698.9 | < 0.001 |
| Never Smoked | 60,439 | 56.1 | 1,719 | 87.9 | 58,720 | 55.5 | | |
| Former Smoker | 29,756 | 25.5 | 140 | 6.8 | 29,616 | 25.9 | | |
| Current Smoker | 20,330 | 17.9 | 119 | 5.1 | 20,211 | 18.1 | | |
| **Alcohol Use** | | | | | | | 965.8 | < 0.001 |
| Abstainer | 17,158 | 15.7 | 959 | 50.3 | 16,199 | 15.1 | | |
| Former Drinker | 17,574 | 14.2 | 97 | 5.4 | 17,477 | 14.4 | | |

*(Continued)*

**Table 1.** (Continued)

| ALL | Total | | Asian Indians | | Non-Hispanic Whites | | | |
|---|---|---|---|---|---|---|---|---|
| | N | Wt % | N | Wt % | N | Wt % | Chi-sq | Prob [b] |
| | **111,058** | **100.0** | **1,986** | **100.0** | **109,072** | **100.0** | | |
| Current Use | 74,817 | 68.6 | 903 | 42.6 | 73,914 | 69.1 | | |
| **Physical Activity** | | | | | | | 28.4 | < 0.001 |
| Daily | 7,570 | 6.9 | 131 | 6.6 | 7,439 | 6.9 | | |
| Weekly | 39,527 | 37.3 | 838 | 40.6 | 38,689 | 37.2 | | |
| Monthly/Year | 60,143 | 52.8 | 994 | 51.4 | 59,149 | 52.9 | | |
| **Region** | | | | | | | 47.4 | < 0.001 |
| Northeast | 20,271 | 19.1 | 450 | 24.4 | 19,821 | 19.0 | | |
| Midwest | 28,973 | 27.4 | 356 | 17.6 | 28,617 | 27.6 | | |
| South | 35,173 | 33.9 | 638 | 31.1 | 34,535 | 34.0 | | |
| West | 26,641 | 19.7 | 542 | 26.8 | 26,099 | 19.5 | | |

Abbreviations: HS: High School, LT: Less than; Prob: Probability; Wt: Weighted; Wid/sep/div: Widowed, Separated, Divorced.

[a] BMI was calculated as weight (kg)/height (m)2 and categorized as follows for NHWs: normal, <25.0; overweight, 25.0–29.9; obese, ≥30.0 & WHO Asian criteria for Asian Indians: normal, <23.0; overweight, 23.0–24.9; obese, ≥25.0.

[b] Statistically significant group differences among Asian Indians and Non-Hispanic whites were evaluated with Rao-Scott chi-square tests.

Unadjusted prevalence of diagnosed DM did not differ between Asian Indians and NHWs (7.5% vs 8.8%; P = .09). However, the age-sex adjusted prevalence of diagnosed DM was significantly higher among Asian Indians (8.3% vs 5.8% for NHWs).

## Association of Asian Indian ethnicity to diagnosed diabetes

Unadjusted odds ratios (UOR) and adjusted odds ratios (AOR) and their associated 95% confidence intervals (CI) from unadjusted and adjusted multivariable logistic regression analyses are summarized in Table 2. Without adjustments for age and sex, Asian Indians were as likely as NHWs to report diagnosed DM (UOR = 0.84, 95% CI: 0.68, 1.03; P = 0.0958). However, age-sex adjusted multivariable logistic regression indicated higher odds of reporting diagnosed DM among Asian Indians compared to NHWs (AOR = 1.39, 95% CI: 1.12, 1.71, P < 0.01). This association remained strong and significant after controlling for variables such as age, sex, education, employment, poverty status, and other health insurance (AOR = 1.68, 95% CI: 1.36, 2.09, P < 0.01). In the fully adjusted model, Asian Indians were 1.5 times as likely as NHWs to report DM (AOR = 1.47, 95% CI: 1.16, 1.85, P < 0.05).

A noteworthy finding was the protective effect of college education and high income on diagnosed DM. Those with less than college education were more likely to be diagnosed with diabetes compared to those with college education (less than high school AOR = 1.29; 95% CI: 1.16, 1.42; high school education AOR = 1.22; 95% CI: 1.22, 1.32; some college education AOR = 1.18; 95% CI: 1.10, 1.28. In addition, adults with income 100% below the federal poverty level were more likely to have diagnosed DM (AOR = 1.51; 95% CI: 1.34, 1.70) compared to those with 400% above federal poverty level.

## Association of race/ethnicity and diagnosed diabetes mellitus

We conducted a secondary analysis of stratified logistic regressions by race/ethnicity on diagnosed DM status. These logistic regressions revealed similar factors (sex, age, hypertension,

**Table 2. Associations of adult Asian Indian ethnicity with diagnosed diabetes mellitus (n = 111,058), National Health Interview Survey, 2012–2016 [a].**

| | UOR | 95% CI | Prob | |
|---|---|---|---|---|
| **Model 1—Unadjusted** | | | | |
| **Race/Ethnicity** | | | | |
| Asian Indians | 0.84 | [0.68, 1.03] | 0.0958 | |
| NHW (Reference Group) | | | | |
| | AOR | 95% CI | Prob | |
| **Model 2—Adjusted for age and sex** | | | | |
| **Race/Ethnicity** | | | | |
| Asian Indians | 1.39 | [1.12, 1.71] | 0.0024 | ** |
| NHW (Reference Group) | | | | |
| **Model 3—Adjusted for age, sex, education, employment, poverty status, and health insurance** | | | | |
| **Race/Ethnicity** | | | | |
| Asian Indians | 1.68 | [1.36, 2.09] | < 0.001 | *** |
| NHW (Reference Group) | | | | |
| **Model 4—Adjusted for age, sex, education, employment, poverty status, health insurance, marital status, hypertension, high cholesterol, smoking, physical activity and smoking** | | | | |
| **Race/Ethnicity** | | | | |
| Asian Indians | 1.47 | [1.16, 1.85] | 0.0013 | ** |
| NHW (Reference Group) | | | | |

Abbreviations: UOR: Unadjusted Odds Ratio; AOR, Adjusted Odds Ratio; CI, confidence interval; NHW, Non-Hispanic Whites.

[a] Models were adjusted for age, sex, education, employment, poverty status, health insurance, marital status, hypertension, high cholesterol, smoking, physical activity and smoking. For variables with missing data (education, poverty status, health insurance, body mass index, physical activity, smoking status and alcohol use), missing indicators were used.

heart disease, cholesterol, and marital status) among both AIs and NHWs. While higher income and education were significantly associated with decreased odds of diagnosed DM among NHWs, they were not a protective factor for Asian Indians. Further, obese Asian Indians were 2 times as likely to report diagnosed DM as those with normal BMI (AOR = 1.85, 95% CI: 1.06, 3.24).

## Discussion

The overall aim of this study was to estimated age-sex adjusted prevalence of diagnosed DM, using disaggregated NHIS race/ethnicity data, among Asian Indians living in the US. Results indicated diagnosed DM was 8.3% and higher than earlier NHIS years of 1997–2000 (6.7%) [21] that did not adjust for age and sex. However, the observed DM prevalence among Asian Indians is much lower than 14% as reported in the DIA study (2010) [22] and 38% age-sex adjusted rate of Asian Indian participants living in Chicago and San Francisco aged 40–84 years of the MASALA (Mediators of Atherosclerosis in South Asians Living in America) study [37]. The significantly higher rates in the DIA study can be explained by the diagnosed and undiagnosed DM used for the prevalence rate; the MASALA study used purposive sampling, older age and participants limited to a few counties in the greater Chicago and San Francisco area. The estimated diagnosed DM rate in this study is also lower than the 12.6% prevalence reported by the National Diabetes Statistics Report (2017–2018) that used multiple data

sources. Variation in prevalence rates may also be explained for differing time-period of the data sources (2017–2018 vs 2012–2016) for the National Diabetes Statistics report and the current study as well as the use of non-institutionalized civilian households for estimating diagnosed diabetes [38].

Studies have reported that Asian Indians have higher insulin resistance due to an inherent genetic predisposition, and an increased disease incidence at a younger age and lower body mass index (BMI) [4, 5] Although an estimated 77 million adults in India lived with diagnosed DM in 2019 [39], findings from a cross-sectional multi-level analysis and other research show Asian Indians with a favorable socio-economic profile (i.e. high income and high levels of education) are more likely to report DM compared to other ethnic groups [6–9]. In other words, obesity and sedentary lifestyle predisposes native Indians in India with higher socio-economic status for diabetes [6]. This is contrary to lower risk of DM documented among higher socio-economic status strata in the United States [40].

Asian Indians in the current study had higher socio economic status in the United States, and is consistent with a prior NHIS study [21]. Higher level of socioeconomic status was associated with lower diagnosed DM among Asian Indians. Therefore, life-style factors such as westernized diet, physical inactivity and obesity may play a significant greater role in increasing the risk of DM among Asian Indians in the United States. Raising awareness among the population regarding the deleterious effects of a high-fat, high-carbohydrate, high-calorie diet and encouraging them to continue the more healthy traditional foods could help individuals make healthy dietary choices, helping to reduce the risk of not only type 2 diabetes but related comorbid chronic conditions as well.

Asian Indians had lower rates of smoking and higher levels of exercise as compared to NHWs, which concurs with published studies [21]. Using data estimates from the National Survey on Drug Use and Health, it has been found that, compared to 14 racial/ethnic groups, Asian Indians had one of the lowest rates of tobacco use [41]. Furthermore, obesity increased the odds of a diagnosed DM in both groups. However, estimated obesity rates among Asian Indians were lower than NHWs when the standard criteria were used and elevated with the WHO Asian criteria. Similar observations were also made in the DIA study that reported an obesity rate of 11 and 49.8% using the standard and WHO Asian criteria, respectively [22]. Other studies have also reported similar trends [21, 23].

Studies have documented that Asian Indians may be susceptible to central obesity and higher body fat percentage for the same BMI, in comparison to other groups [42, 43] There is also evidence that South Asians often perceive themselves to be normal weight and underestimate their risk for chronic diseases even when they have higher BMI [44]. Acculturation and adaptation of westernized culture has shown to result in changes to ethnic dietary habits, e.g., an increased consumption of refined grains, high energy dense food and sugary drinks and beverages leading to high glycemic load that increase the risk for diabetes [45]. In addition, health outcomes of immigrant tend to decline with longer duration of stay in the US and Asian Indians who do not prioritize preventive health/wellness checks, health diet and exercise or physical activity may be at higher risk for diabetes. Since access to healthcare and English language proficiency is not a barrier for health promotion/ lifestyle interventions, tailored messages to improve awareness of disease risk, health beliefs or perceptions and support for behavior modification can be recommended by healthcare providers for both primary and secondary prevention, especially in 1st generation Asian Indians due to an emphasis on careers and family priorities. These findings suggest that culturally appropriate diabetes prevention programs, shown to be efficacious in preventing diabetes among Asian Indians should be widely disseminated [46, 47].

### Strengths and limitations

The strength of the current study includes is the utilization of disaggregated national data for Asian Indians for multiple years. Use of WHO Asian criteria for BMI cut point for Asian Indian and adjustment of a comprehensive list of risk factors in the multivariate model are additional strengths. However, the limitations include its cross-sectional design, a small sample size despite the aggregation of multiple years that did not allow assessment of heterogeneity among the Asian Indian subgroup. Also, our definition of diabetes included only self-reporting of diagnosed diabetes and may underestimate the true prevalence of diabetes in both groups. For example, some persons may have been classified as not having diabetes when in fact they had undiagnosed diabetes; as much as one-quarter to more than one-third of all diabetes may be undiagnosed. Unfortunately, the NHIS data did not allow us to capture undiagnosed diabetes. However, additional analysis on diagnosed diabetes, pre-or borderline diabetes, and no diabetes between the two groups showed no systematic differences in the rates (P = .224). In addition, there was a lack of cross-cultural validity of the physical activity measurement. More specifically, recall bias of specific vigorous physical activities, the description of exercises would vary according to different ethnic groups. Lastly, important confounders such as clinical information, dietary habits, or family history could potentially affect the interpretation and association of Asian Indian ethnicity with diagnosed DM. Family history of DM is a significant risk factor that could not be adjusted as the data was available only in 2016.

## Conclusion

In conclusion, the current study confirmed the higher prevalence of diagnosed DM among Asian Indians as compared to NHWs, despite favorable socio-economic profiles and low smoking rates. The study's findings also highlighted that among Asian Indians obesity rates were high and linked with diabetes. Findings demonstrate the need for health education and culturally tailored diabetes prevention programs that are critical in preventing DM among Asian Indians living in the US.

## New findings

- This study demonstrates its strength by estimating the prevalence of DM using a disaggregated nationally represented data for Asian Indians using the WHO Asian criteria.

- Using the WHO criteria, our study estimated that obesity rate was significantly higher i.e., 46.1% for Asian Indians.

- Age-sex adjusted prevalence of diagnosed DM was significantly higher among Asian Indians (8.3% vs 5.8% for NHWs).

- Asian Indians who do not prioritize preventive health/wellness checks, health diet and exercise or physical activity may be at higher risk for diabetes.

## Integrity statement

The primary author, Ranjita Misra is the guarantor of this work and takes responsibility for the integrity of the data and the accuracy of the data analysis. The article in the current form is approved by all authors and confirms to the criteria by the International Committee for Medical Journal Editors (ICMJE).

## Prior presentation information

The study was presented at the American Public Health Association Annual Conference.

## Author Contributions

**Conceptualization:** Ranjita Misra, Usha Sambamoorthi.

**Formal analysis:** Usha Sambamoorthi.

**Methodology:** Usha Sambamoorthi.

**Project administration:** Trupti Dhumal.

**Writing – original draft:** Ranjita Misra, Usha Sambamoorthi.

**Writing – review & editing:** Ranjita Misra, Suresh S. Madhavan, Trupti Dhumal.

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
