## [Decision Letter · Decision Letter 0]

6 Dec 2022

PGPH-D-22-01699

Prevalence and Factors Associated with Diagnosed Diabetes Mellitus among Asian Indian Adults in the United States

Dear Dr. Misra,

Thank you for submitting your manuscript to PLOS Global Public Health. After careful consideration, we feel that it has merit but does not fully meet PLOS Global Public Health’s publication criteria as it currently stands. Therefore, we invite you to submit a revised version of the manuscript that addresses the points raised during the review process.

We look forward to receiving your revised manuscript.

Kind regards,

Ransome Eke, M.D., Ph.D., MPH

Academic Editor

Journal Requirements:

1. Please note that your Data Availability Statement is currently missing a link to access each database. If your manuscript is accepted for publication, you will be asked to provide these details on a very short timeline. We therefore suggest that you provide this information now, though we will not hold up the peer review process if you are unable.

2. Please insert an Ethics Statement at the beginning of your Methods section, under a subheading 'Ethics Statement'. It must include:

1) The name(s) of the Institutional Review Board(s) or Ethics Committee(s)

2) The approval number(s), or a statement that approval was granted by the named board(s) 

[3) (for human participants/donors) - A statement that formal consent was obtained (must state whether verbal/written) OR the reason consent was not obtained (e.g. anonymity). NOTE: If child participants, the statement must declare that formal consent was obtained from the parent/guardian.]

Additional Editor Comments (if provided):

Reviewers' comments:

Reviewer's Responses to Questions

**Comments to the Author**

1. Does this manuscript meet PLOS Global Public Health’s publication criteria? Is the manuscript technically sound, and do the data support the conclusions? The manuscript must describe methodologically and ethically rigorous research with conclusions that are appropriately drawn based on the data presented.

Reviewer #1: Yes

2. Has the statistical analysis been performed appropriately and rigorously?

Reviewer #1: Yes

3. Have the authors made all data underlying the findings in their manuscript fully available (please refer to the Data Availability Statement at the start of the manuscript PDF file)?

Reviewer #1: Yes

4. Is the manuscript presented in an intelligible fashion and written in standard English?

Reviewer #1: Yes

5. Review Comments to the Author

Reviewer #1: This study seems like a great initiative to increase awareness about NCD among south Asians. I have a few concerns here.

1. Self reporting of diagnosed DM cases without any investigations will potentially create a bias. Asymptomatic early DM cases have been excluded/missed here. Is there any investigation report in their personal file?

2. Family history of DM- an important risk factor has not been adjusted.

3. 60,020 women have been included in the study. Is there any history of GDM? Have you excluded GDM samples from the study? If so ,please mention that.

4. How did you confirm high cholesterol?

6. PLOS authors have the option to publish the peer review history of their article (what does this mean?). If published, this will include your full peer review and any attached files.

**Do you want your identity to be public for this peer review?** For information about this choice, including consent withdrawal, please see our Privacy Policy.

Reviewer #1: No

---

## [Editor Report · Decision Letter 1]

11 Jan 2023

Prevalence and Factors Associated with Diagnosed Diabetes Mellitus among Asian Indian Adults in the United States

PGPH-D-22-01699R1

Dear Dr. Misra,

We are pleased to inform you that your manuscript 'Prevalence and Factors Associated with Diagnosed Diabetes Mellitus among Asian Indian Adults in the United States' has been provisionally accepted for publication in PLOS Global Public Health.

Best regards,

Ransome Eke, M.D., Ph.D., MPH

Academic Editor